# Development of insomnia in patients with stroke: A systematic review and meta-analysis

**Junwei Yang**[1⊛‡]*, **Aitao Lin**[2‡], **Qingjing Tan**[1‡], **Weihua Dou**[1], **Jinyu Wu**[1], **Yang Zhang**[1], **Haohai Lin**[1], **Baoping Wei**[1], **Jiemin Huang**[1], **Juanjuan Xie**[1⊛]*

1 The First Affiliated Hospital of Guangxi University of Chinese Medicine, Nanning, Guangxi, 530023, China,
2 Guangxi University of Traditional Chinese Medicine, Nanning, Guangxi, 530001, China

⊛ These authors contributed equally to this work.
‡ JY, AL and QT are share first authorship on this work
* 404121650@qq.com (JY); 398601398@qq.com (JX)

## Abstract

### Background and aim

Stroke is a serious threat to human life and health, and post-stroke insomnia is one of the common complications severely impairing patients' quality of life and delaying recovery. Early understanding of the relationship between stroke and post-stroke insomnia can provide clinical evidence for preventing and treating post-stroke insomnia. This study was to investigate the prevalence of insomnia in patients with stroke.

### Methods

The Web of Science, PubMed, Embase, and Cochrane Library databases were used to obtain the eligible studies until June 2023. The quality assessment was performed to extract valid data for meta-analysis.

The prevalence rates were used a random-efect. $I^2$ statistics were used to assess the heterogeneity of the studies.

### Results

1. Twenty-six studies met the inclusion criteria for meta-analysis, with 1,193,659 participants, of which 497,124 were patients with stroke.

2. The meta-analysis indicated that 150,181 patients with stroke developed insomnia during follow-up [46.98%, 95% confidence interval (CI): 36.91–57.18] and 1806 patients with ischemic stroke (IS) or transient ischemic attack (TIA) developed insomnia (47.21%, 95% CI: 34.26–60.36). Notably, 41.51% of patients with the prevalence of nonclassified stroke developed insomnia (95% CI: 28.86–54.75). The incidence of insomnia was significantly higher in patients with acute strokes than in patients with nonacute strokes (59.16% vs 44.07%, $P < 0.0001$).

3. Similarly, the incidence of insomnia was significantly higher in the patients with stroke at a mean age of ≥65 than patients with stroke at a mean age of <65 years (47.18% vs 40.50%, $P$

**Data Availability Statement:** All relevant data are within the manuscript and its Supporting Information files.

**Funding:** This work was supported by Administration of Traditional Chinese medicine in guangxi, self-financing scientific research subject [grant numbers GXZYA20220072]; Natural Science Foundation of Guangxi[grant numbers 2023GXNSFAA026200]; Hospital scientific research project of the First Affiliated Hospital of Guangxi University of Traditional Chinese Medicine [grant numbers 2021QN008]; Guangxi University of Traditional Chinese Medicine research project [grant numbers 2022QN019]. This work was supported by Junwei YANG and Qingjing TAN.

**Competing interests:** The authors have declared that no competing interests exist.

$< 0.05$). Fifteen studies reported the follow-up time. The incidence of insomnia was significantly higher in the follow-up for $\geq 3$ years than follow-up for $<3$ years (58.06% vs 43.83%, $P < 0.05$). Twenty-one studies used the Insomnia Assessment Diagnostic Tool, and the rate of insomnia in patients with stroke was 49.31% (95% CI: 38.59–60.06). Five studies used self-reporting, that the rate of insomnia in patients with stroke was 37.58% (95% CI: 13.44–65.63).

## Conclusions

Stroke may be a predisposing factor for insomnia. Insomnia is more likely to occur in acute-phase stroke, and the prevalence of insomnia increases with patient age and follow-up time. Further, the rate of insomnia is higher in patients with stroke who use the Insomnia Assessment Diagnostic Tool.

## 1 Introduction

Stroke is the second most morbid and deadly disease globally, which is characterized by high morbidity, disability, mortality, and recurrence. It substantially threatens human life, health, and quality of life [1,2]. Previous study revealed that neuropsychiatric disorders frequently affect stroke survivors, such as insomnia, depression, or anxiety and so on [3]. Similarly, One third of stroke patients met the diagnostic criteria of insomnia, and patients may experience difficulty falling asleep, difficulty with sleep persistence, and early awakening [4].

Insomnia is the most common sleep disorder prevalent in people of all ages. In severe cases, it can affect daytime work and life, and even cause emotional disorders [5]. The incidence of insomnia increases with the increase in social pressure [6]. Study showed that the incidence of insomnia in stroke patients is higher than the normal healthy population, and some patients with insomnia may be more prone to stroke risk [7]. As increasing studies showed that insomnia has a bidirectional relationship with stroke, which may be an independent risk factor for stroke. Further, stroke may also be a predisposing factor for insomnia [8]. Therefore, it is essential to understand the relationship between stroke and post-stroke insomnia in an early stage to provide a clinical basis for the early prevention and treatment of post-stroke insomnia. The study aimed to investigate the prevalence of insomnia in patients with stroke.

## 2 Research design and method

The study was conducted and designed in strict accordance with the Preferred Reporting Items for Systematic Reviews and Meta-Analyses (PRISMA) guidelines [9,10].

### 2.1 Data source and selection process

Literature related to the occurrence of developmental insomnia in stroke patients was collected through PubMed, The Cochrane Library, Web of Science, and Embase databases until June 2023.

### 2.2 Search strategy

We searched the related literature by the subject terms, such as "Stroke", "Cerebrovascular Accident", "Insomnia", "Insomnia Disorder", etc. The following search strategy for the PubMed database (Fig 1).

#1    Stroke[Mesh]

#2    Stroke [Mesh]) OR (stroke[Title/Abstract])) OR (Strokes[Title/Abstract])) OR (Cerebrovascular Accident[Title/Abstract])) OR (Cerebrovascular Accidents[Title/Abstract])) OR (CVA (Cerebrovascular Accident[Title/Abstract]))) OR (CVAs (Cerebrovascular Accident[Title/Abstract]))) OR (Cerebrovascular Apoplexy[Title/Abstract])) OR (Apoplexy, Cerebrovascular[Title/Abstract])) OR (Vascular Accident, Brain[Title/Abstract])) OR (Brain Vascular Accident[Title/Abstract])) OR (Brain Vascular Accidents[Title/Abstract])) OR (Brain Vascular Accidents[Title/Abstract])) OR (Cerebrovascular Stroke[Title/Abstract])) OR (Cerebrovascular Strokes[Title/Abstract])) OR (Stroke, Cerebrovascular[Title/Abstract])) OR (Strokes, Cerebrovascular[Title/Abstract])) OR (Apoplexy[Title/Abstract])) OR (Cerebral Stroke[Title/Abstract])) OR (Cerebral Strokes[Title/Abstract])) OR (Stroke, Cerebral[Title/Abstract])) OR (Strokes, Cerebral[Title/Abstract])) OR (Stroke, Acute[Title/Abstract])) OR (Acute Stroke[Title/Abstract])) OR (Acute Strokes[Title/Abstract])) OR (Strokes, Acute[Title/Abstract])) OR (Strokes, Acute[Title/Abstract])) OR (Acute Cerebrovascular Accident[Title/Abstract])) OR (Acute Cerebrovascular Accidents[Title/Abstract])) OR (Cerebrovascular Accidents, Acute[Title/Abstract])

#3 insomnia [Mesh]

#4 insomnia [Mesh] OR insomnia [Title/Abstract]) OR (Disorders of Initiating[Title/Abstract] AND Maintaining Sleep[Title/Abstract])) OR (DIMS (Disorders of Initiating[Title/Abstract] AND Maintaining Sleep[Title/Abstract]))) OR (Early Awakening[Title/Abstract])) OR (Awakening, Early Nonorganic Insomnia[Title/Abstract])) OR (Insomnia, Nonorganic[Title/Abstract])) OR (Primary Insomnia[Title/Abstract])) OR (Insomnia, Primary[Title/Abstract])) OR (Transient Insomnia[Title/Abstract])) OR (Insomnia, Transient[Title/Abstract])) OR (Rebound Insomnia[Title/Abstract])) OR (Insomnia, Rebound[Title/Abstract])) OR (Secondary Insomnia[Title/Abstract])) OR (Insomnia, Secondary[Title/Abstract])) OR (Sleep Initiation Dysfunction[Title/Abstract])) OR (Dysfunction, Sleep Initiation[Title/Abstract])) OR (Sleeplessness[Title/Abstract])) OR (Insomnia Disorder[Title/Abstract])) OR (Chronic Insomnia[Title/Abstract])) OR (Psychophysiological Insomnia[Title/Abstract]

#5 #2 AND #4

**Fig 1. Search strategy of PubMed.**

## 2.3 Eligibility criteria and study selection

In the study, we included the cohort studies and cross-sectional studies about stroke patients who developed insomnia in English language. Stroke patients met the diagnostic criteria of the Essentials of Diagnosis of Various Cerebrovascular Diseases [11]. Insomnia patients were diagnosed through recognized assessment tools such as the Pittsburgh Sleep Quality Index (PSQI), Hamilton Depression Scale (HDS), or self-reported symptoms of insomnia and met the diagnostic criteria of the American Academy of Sleep in 2014 [12]. We excluded the duplicate records, case reports, reviews and so on.

## 2.4 Exclusion criteria

We excluded the duplicate literature, case reports, reviews and the the literature with incomplete data indicators, or the information was not available.

## 2.5 Data extraction

**2.5.1 Literature screening and information extraction.** LAT and TQJ screened the included literatures. The extracted information mainly included the basic information of the literatures: first author name, the time of publication, sample size, the country, the follow-up time and the number of positive cases. In case of disagreement between two researchers in the literature screening or data extraction process, the decision was submitted to the third researcher (YJW).

**2.5.2 Literature quality assessment.** The methodological quality of the included studies was assessed using the Critical Appraisal Tool for Prevalence Studies [13,14]. Any disagreements by the researchers were submitted to a third researcher (YJW).

## 2.6 Statistical analysis

In the study, we used systematic Meta-Analysis software version 3 to calculate the statistical analyse [15]. The fixed effects model was used in $P \geq 0.10$ and $I^2 \leq 50\%$, and random-effects model was used in $P < 0.10$ and/or $I^2 > 50\%$, which was necessary to find the source of heterogeneity and perform subgroup analysis or sensitivity analysis [16–18].

# 3 Results

## 3.1 PROSPERO registration

Registration number: CRD42023452419.

## 3.2 Literature search results

We got 1507 literatures from databases, of which 469 were duplicates and hence excluded. Further, we excluded 927 studies by the exclusion criteria. Overall, 111 studies were retained for the full-text evaluation, and finally 26 studies were included in the meta-analysis (Fig 2) [7,19–43].

## 3.4 Basic characteristics of the included studies

The 26 included studies (13 prospective cohort studies, 10 cross-sectional studies, 2 retrospective studies, and 1 multicenter observational study) were published between 2002 and 2023. Overall, the 26 studies included had 1,193,659 participants, of which 497,124 were patients with stroke. The details are shown in Table 1.

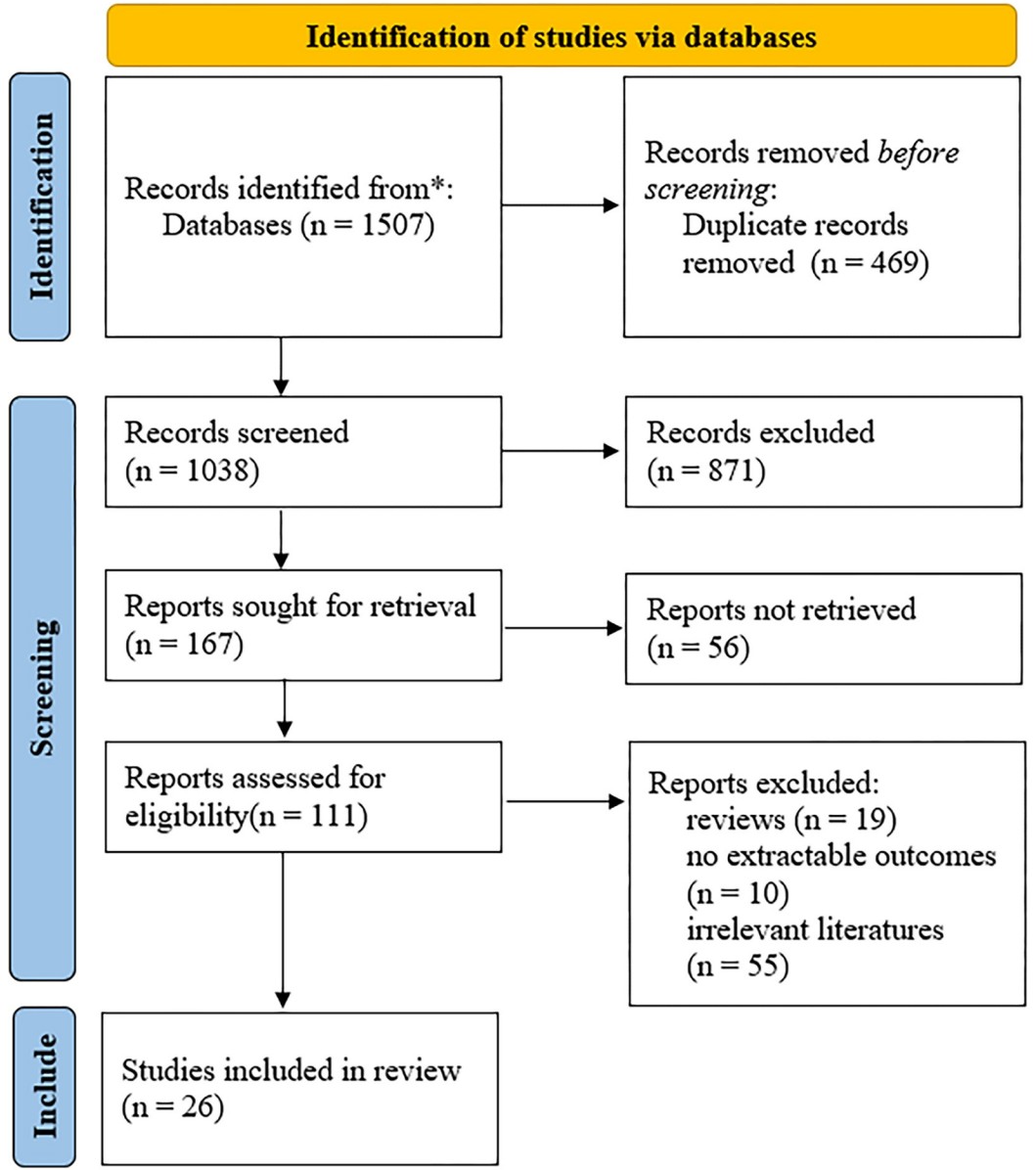

**Fig 2. Literature screening process and results.**

### 3.5 Quality of included studies

Table 2 shows the quality assessment of the included studies. 69.23% (eighteen studies) studies determined the prevalence of insomnia in stroke patients in a sufficient sample size. To assess insomnia, most studies used standardized instruments or validated diagnostic criteria (80.77%). The details are shown in Table 2.

**Table 1. Study characteristics.**

| First author/ Date | Country | Years | Total N (%, n of stroke) | Design | Stroke/ n | Insomnia/ n | Followup period | Mean Age, years (SD) | Stroke type | Gender % male |
|---|---|---|---|---|---|---|---|---|---|---|
| Simone B. Duss et al 2023[19] | Bern | 2015.11–2016.7 | 437 (100%) | Prospective cohort study | 437 | 168 | 2 yrs | 65.1 ± 13.0 | IS、TIA | 63.8% |
| Won-Hyoung Kim et al 2017 [20] | Korean | 2014.3–2015.12 | 214 (100%) | Cohort study | 214 | 128 | 1 mo | NR | IS、Hemorrhagic stroke | NR |
| Hye-Mi Moon et al 2019[21] | Korean | 2010–2012, 2013 | 504 (100%) | Cross-sectional Population based survey | 504 | 123 | NR | 64.4 ± 0.7 | stroke | 55.7% |
| A. Leppävuori et al 2002[7] | NR | NR | 277 (100%) | Cross-sectional interview | 277 | 157 | 3 mo | 70.7 ± 7.5 | IS | 50.9% |
| Ruo-lin Zhu et al 2022[22] | China | 2020.1–2021.5 | 94 (100%) | Cross-sectional survey | 94 | 59 | 16 mo | NR | IS | 70.2% |
| A. Katharina Helbig et al 2015[23] | Germany | NR | 15746 (5.8%, n = 917) | Cross-sectional survey | 917 | 769 | 14 yrs (md) | NR | stroke | 62.16% |
| Azizi A Seixas et al 2019[24] | USA | 2000–2015 | 1108043 (43.9%, n = 486619) | Cross-sectional Population based survey | 486619 | 145207 | NR | 47.5 ± 14.15 | stroke | 47.3% |
| Biljana Kojic et al 2021[25] | Tuzla | NR | 110 (100%) | Prospective study | 100 | 100 | NR | 65.13 ± 9.27 | Stroke | 59% |
| Chien-Yi Hsu et al 2015[26] | Taiwan, China | NR | 44080 (2.38%, n = 1049) | Cross-sectional cohort | 1049 | 743 | 10 yrs | NR | Stroke | NR |
| Faizul Hasan et al 2023[27] | Taiwan, China | 2004.1–2017.9 | 1775 (100%) | Retrospective Cohort Study | 1775 | 146 | NR | 67.6 ± 14.91 | Stroke | 58.6% |
| Gul M C et al 2016[28] | Turkey | NR | 81 (100%) | Retrospective study | 81 | 30 | 5 yrs | 66.5 ± 10.3 | IS | 50.6% |
| Hui-Ju Tsai et al 2022[29] | Taiwan, China | 2020.7–2021.10 | 195 (100%) | Prospective study | 195 | 58 | 15 mo | 64.1 ± 8.9 | IS | 59.5% |
| Ipek Sonmez et al 2019[30] | Famagusta | 2016.1–2017.2 | 55 (100%) | Cross-sectional observational study | 55 | 32 | NR | 69 ± 11 | Stroke | NR |
| Jinil Kim et al 2015[31] | China | 2013.10–2014.6 | 80 (100%) | Prospective study | 80 | 57 | NR | 63.8 ± 13.6 | IS、Hemorrhagic stroke | 67.5% |
| Keun T K et al.2017[32] | Korean | NR | 241 (100%) | Prospective study | 241 | 108 | NR | 64.2 ± 11.9 | AIS | 60.6% |
| Kyung-Lim Joa et al 2017[33] | Korean | NR | 208 (100%) | Multicenter-observational and correlation study | 208 | 56 | NR | 61.53 ± 12.58 | Stroke | 54% |
| Li-Jun Li et al 2018[34] | China | 2008.4–2010.4 | 1062 (100%) | Prospective Cohort Study | 1062 | 489 | 6 yrs | 60.47 ± 11.57 | Stroke | 65.7% |
| M. Sieminski et al 2009[35] | Poland | 1995–2005 | 90 (100%) | Prospective study | 90 | 65 | NR | 66.5 ± 12.8 | IS | 46.7% |
| Mayura T I et al 2019[36] | Australia | 2016.8–2018.1 | 104 (100%) | prospective cohort study | 104 | 31 | 17 mo | 76 ± 7 | Stroke | 52.9% |
| Min-Y K et al 2018[37] | Korean | 2010–2014 | 17601 (2%, n = 360) | Cross-sectional survey study | 360 | 170 | 4 yrs | NR | Stroke | NR |
| Nick Glozier et al 2017[38] | Australia | 2008–2010 | 368 (100%) | Prospective cohort study | 368 | 124 | 1 year | NR | IS、Hemorrhagic stroke | 68.2% |
| Wai-Kwong Tang et al 2015 [39] | Hong Kong, China | 2008.6–2011.9 | 336 (100%) | Cross-sectional survey study | 336 | 149 | 3 mo | 66.1 ± 10.2 | Acute stroke | 60.4% |

(*Continued*)

**Table 1.** (Continued)

| First author/ Date | Country | Years | Total N (%, n of stroke) | Design | Stroke/ n | Insomnia/ n | Followup period | Mean Age, years (SD) | Stroke type | Gender % male |
|---|---|---|---|---|---|---|---|---|---|---|
| | | | **Study details** | | | | | **Sample characteristics (stroke sample)** | | |
| Won-Hyoung Kim et al 2019 [40] | Korean | 2016.7–2018.8 | 112 (100%) | Cohort study | 112 | 40 | NR | NR | Stroke | 54.5% |
| Xiao-Wei Fan et al 2022[41] | China | 2015.8–2018.3 | 1619 (100%) | Prospective study | 1619 | 1137 | 3 mo | 60.8 ± 10.7 | AIS or TIA | 72.5% |
| Yitao He et al 2019[42] | China | 2016.1–2018.6 | 152 (100%) | Prospective study | 152 | 24 | 3 mo | 65.25 ± 13.56 | AIS | 67.76% |
| Alia H. Mansour et al 2020[43] | Egypt | 2015.1–2015.12 | 75 (100%) | Cross-sectional prospective study | 75 | 11 | NR | 59.3 ± 5.34 | Stroke | 45.3% |

## 3.6 Meta-analysis

We used the random-effects model to pool prevalence of insomnia in patients with stroke. 150,181 patients with stroke developed insomnia during the follow-up and the pool prevalence was 46.98% (95% CI: 36.91–57.18) (Fig 3).

Moreover, nine studies examined the occurrence of insomnia in patients with IS or TIA. The result showed that the prevalence of insomnia among patients with IS or TIA was 47.21% (95% CI: 34.26–60.36) (Fig 4).

Four studies explicitly examined the prevalence of insomnia among IS or hemorrhage patients and the prevalence was 44.09% (95% CI: 19.84–69.92), while twelve studies did not specify the type of stroke (Fig 5).

Five studies explored the odds of insomnia in patients with acute stroke, and the prevalence was 59.16% (95% CI: 24.18–89.55) (Fig 6). Meanwhile, the odds of insomnia in patients with nonacute stroke was 44.07% in twenty-one studies (95% CI: 34.74–53.61) (Fig 7).

In the subgroup analysis, we found that the incidence of insomnia was significantly higher in the patients with stroke at a mean age of ≥65 than patients with stroke at a mean age of <65 years, which was [47.18% (95% CI: 26.7–68.16) vs 40.50% (95% CI: 26.21–55.66), $P<0.05$] (Figs 8 and 9).

Moreover, concerning the follow-up duration of the participants, we found that the prevalence of insomnia was significantly higher in the follow-up duration was ≥3 years than those with a follow-up period <3 years (58.06% vs 43.83%, $P < 0.001$) (Figs 10 and 11).

In the end, the subgroup analyse was performed based on the use of insomnia assessment diagnostic tools (clinical assessment diagnostic tools vs self-report). Twenty-one studies used insomnia assessment diagnostic tools, and the insomnia rate in stroke patients was 49.31% (95% CI: 38.59–60.06) (Fig 12). Five studies used self-report, and the results indicated that the insomnia rate in stroke patients was 37.58% (95% CI: 13.44–65.63) (Fig 13).

## 4 Discussion

### 4.1 Key findings

This study was an updated review about the prevalence of insomnia among patients with stroke. Further, 26 studies from 11 countries were included, of which 15 studies were conducted in Asia (57.69%) and the remaining studies were conducted outside Asia. Of the 26 included studies, 21 used diagnostic tools and 5 used nondiagnostic tools for assessing insomnia.

**Table 2. Quality of included studies.**

| Study | Response | | | | | | | | | |
|---|---|---|---|---|---|---|---|---|---|---|
| | Q1 | Q2 | Q3 | Q4 | Q5 | Q6 | Q7 | Q8 | Q9 | Q10 |
| Simone B. Duss et al 2023[19] | Y | Y | Y | Y | Y | Y | N | Y | Y | Y |
| Won-Hyoung Kim et al 2017[20] | Y | Y | Y | Y | Y | Y | Y | Y | Y | Y |
| Hye-Mi Moon et al 2019[21] | Y | Y | Y | Y | Y | N | Y | Y | Y | Y |
| A. Leppävuori et al 2002[7] | Y | U | Y | Y | Y | Y | Y | Y | Y | Y |
| Ruo-lin Zhu et al 2022[22] | Y | Y | N | Y | Y | Y | Y | Y | Y | Y |
| K.H. et al 2015[23] | Y | Y | Y | Y | Y | U | U | Y | Y | Y |
| Azizi A Seixas et al 2019[24] | Y | Y | Y | Y | Y | N | Y | Y | Y | Y |
| Biljana Kojic et al 2021[25] | Y | U | Y | Y | Y | Y | Y | Y | Y | Y |
| Chien-Yi Hsu et al 2015[26] | Y | Y | Y | Y | Y | Y | Y | Y | Y | Y |
| Faizul Hasan et al 2023[27] | Y | Y | Y | Y | Y | N | Y | Y | Y | Y |
| Gul M C et al 2016[28] | Y | U | N | Y | Y | Y | Y | Y | Y | Y |
| Hui-Ju Tsai et al 2022[29] | Y | Y | Y | Y | Y | Y | Y | Y | Y | Y |
| Ipek Sonmez et al 2019[30] | Y | Y | Y | Y | Y | Y | U | Y | Y | Y |
| Jinil Kim et al 2015[31] | Y | Y | N | Y | Y | Y | U | Y | Y | Y |
| Keun T K et al.2017[32] | Y | Y | Y | Y | Y | Y | Y | Y | Y | Y |
| Kyung-Lim Joa et al 2017[33] | Y | Y | Y | Y | Y | Y | Y | Y | Y | Y |
| Li-Jun Li et al 2018[34] | Y | U | Y | Y | Y | Y | Y | Y | Y | Y |
| M. Sieminski et al 2009[35] | Y | U | N | U | Y | Y | Y | Y | Y | Y |
| Mayura T I et al 2019[36] | Y | U | N | Y | Y | Y | U | Y | Y | Y |
| Min-Y K et al 2018[37] | Y | Y | Y | Y | Y | N | N | Y | Y | Y |
| Nick Glozier et al 2017[38] | Y | Y | Y | Y | Y | Y | N | Y | Y | Y |
| Wai-Kwong Tang et al 2015[39] | Y | U | Y | Y | Y | Y | Y | Y | Y | Y |
| Won-Hyoung Kim et al 2019[40] | Y | U | N | Y | Y | Y | Y | Y | Y | Y |
| Xiao-Wei Fan et al 2022[41] | Y | Y | Y | Y | Y | Y | Y | Y | Y | Y |
| Yitao He et al 2019[42] | Y | Y | N | Y | Y | Y | Y | Y | Y | Y |

(*Continued*)

**Table 2.** (Continued)

| Study | Response | | | | | | | | | |
|---|---|---|---|---|---|---|---|---|---|---|
| | Q1 | Q2 | Q3 | Q4 | Q5 | Q6 | Q7 | Q8 | Q9 | Q10 |
| Alia H. Mansour et al 2020[43] | Y | Y | N | Y | Y | Y | Y | Y | Y | Y |

Keys

Q1-Q10 represents questions used to assess the quality of included studies, which are listed below.

Q1. Sample frame appropriate to address the target population.

Q2. Appropriate sampling of study participants.

Q3. Adequate sample size.

Q4. Study subjects and setting described in detail.

Q5. Data analysis conducted with sufficient coverage of the identified sample.

Q6. Valid methods used for the identification of insomnia or insomnia symptoms.

Q7. Valid methods used for the identification of stroke.

Q8. Condition measured in a standard, reliable way for all participants.

Q9. Appropriate statistical analysis.

Q10. Adequate response rate, if not, was low response rate managed appropriately.

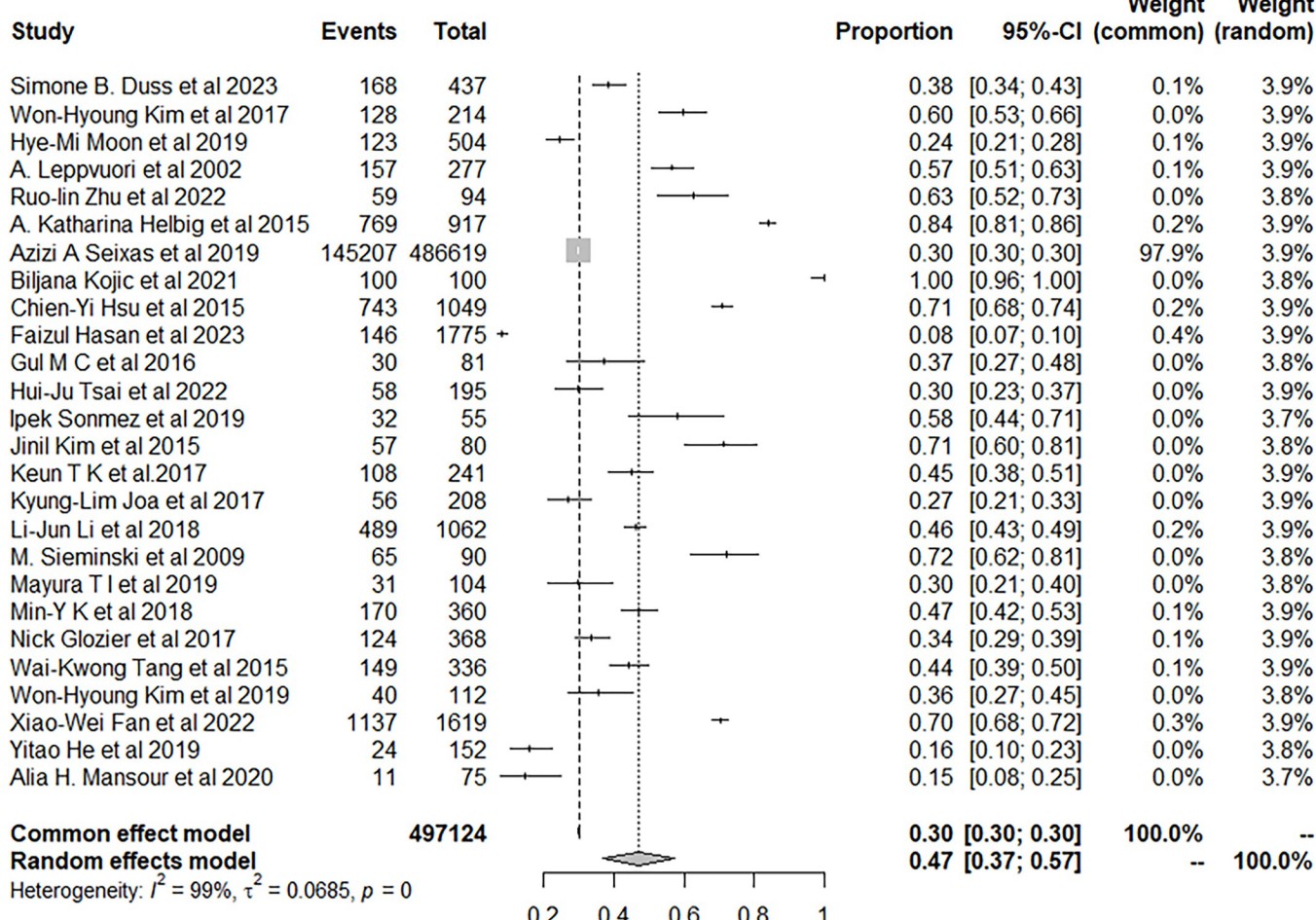

**Fig 3. Forest plot of the meta-analysis of prevalence of insomnia among stroke patients.**

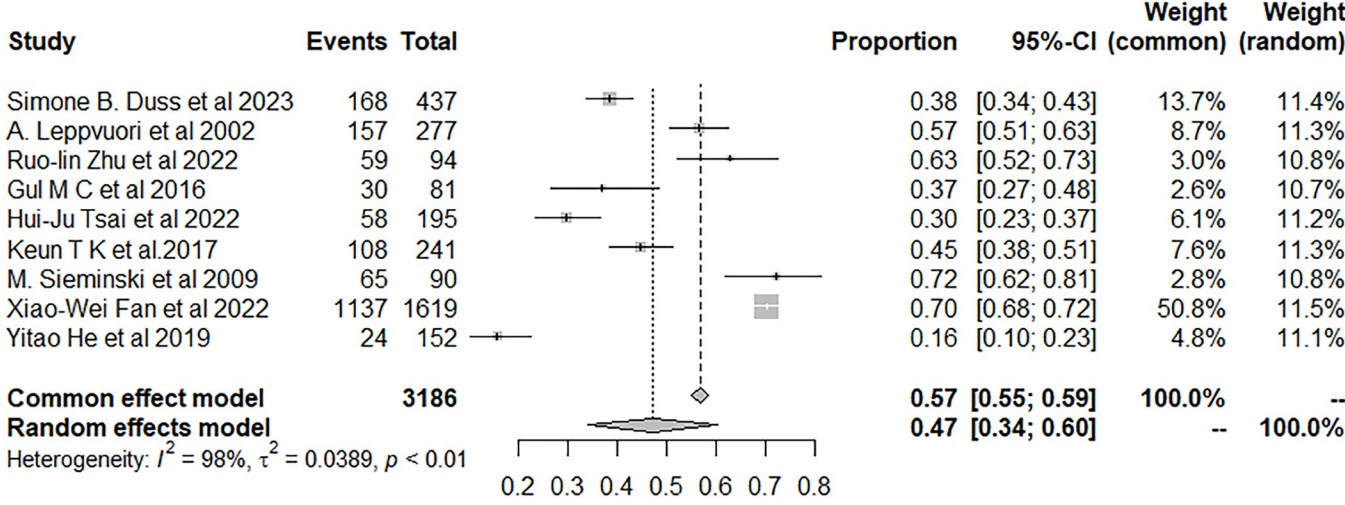

**Fig 4. Forest plot of the meta-analysis of prevalence of insomnia among IS or TIA patients.**

Overall, our meta-analysis indicated that the rate of insomnia after stroke was 48.37%. It was estimated that incidence of IS or TIA (47.21%) was higher than that of unclassified stroke (41.51%); the rate of acute-phase stroke was higher (59.16%) than that of nonacute-phase stroke (36.31%); the proportion of patients with a mean age ≥65 years was higher (47.18%) than the proportion of those with a mean age <65 years (44.43%); the duration of follow-up ≥3 years (58.06%) was higher than the duration of follow-up <3 years (43.83%); and the rate of using a diagnostic tool for insomnia assessment was higher (51.16%) than the rate of using a nondiagnostic tool (37.58%). This suggested that post-stroke insomnia was a substantial global public health problem in patients with stroke who needed urgent attention for prevention and treatment.

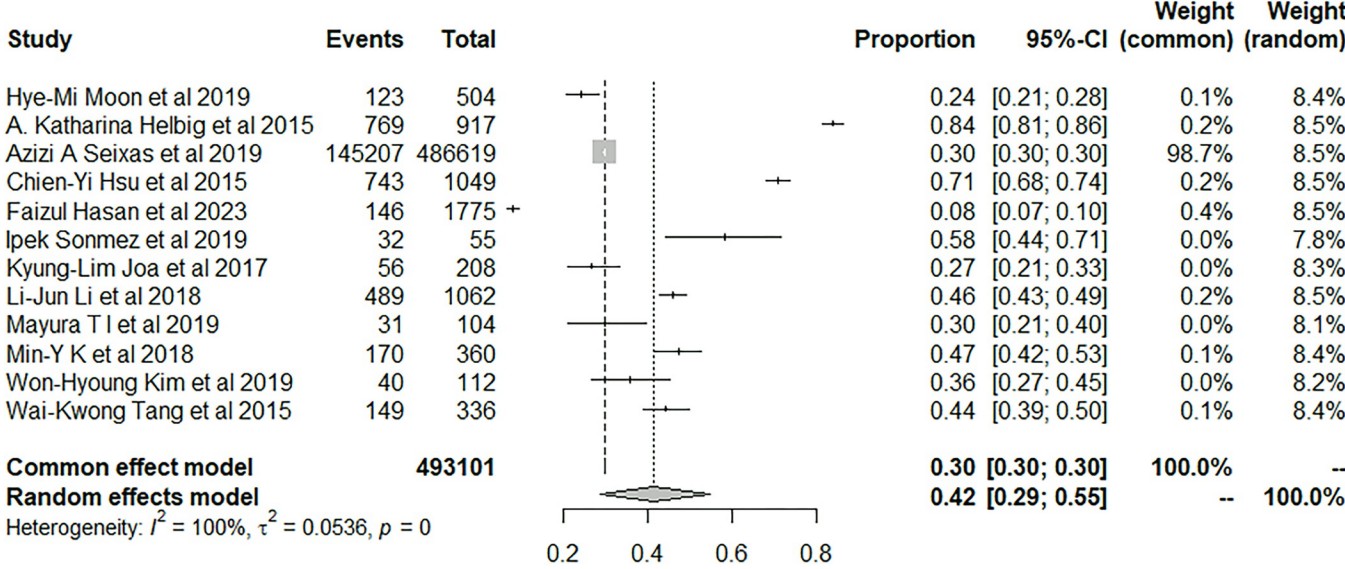

**Fig 5. Forest plot of the meta-analysis of prevalence of insomnia among nonclassified stroke patients.**

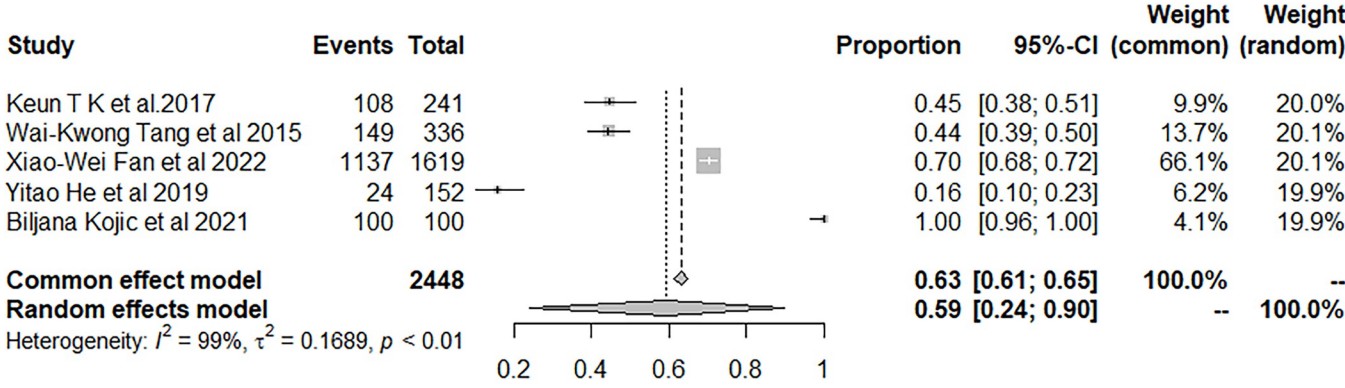

**Fig 6. Forest plot of the meta-analysis of prevalence of insomnia among acute stroke stroke patients.**

## 4.2 Comparisons of the study findings with the available evidence

Our study found that the rate of insomnia after stroke (48.37%) was 1.27 times higher compared with the prevalence in the meta-analysis by Baylan et al. in 2019 (38.2%) [44]. It indicated that the prevalence of post-stroke insomnia continued to increase yearly, and insomnia had a significant negative impact on patients. The data in this study indicated that sleep-related apnea was significantly associated with stroke, and obstructive apnea syndrome might increase the risk of stroke twice [1]. A 4-year follow-up study in Taiwan, China revealed that compared

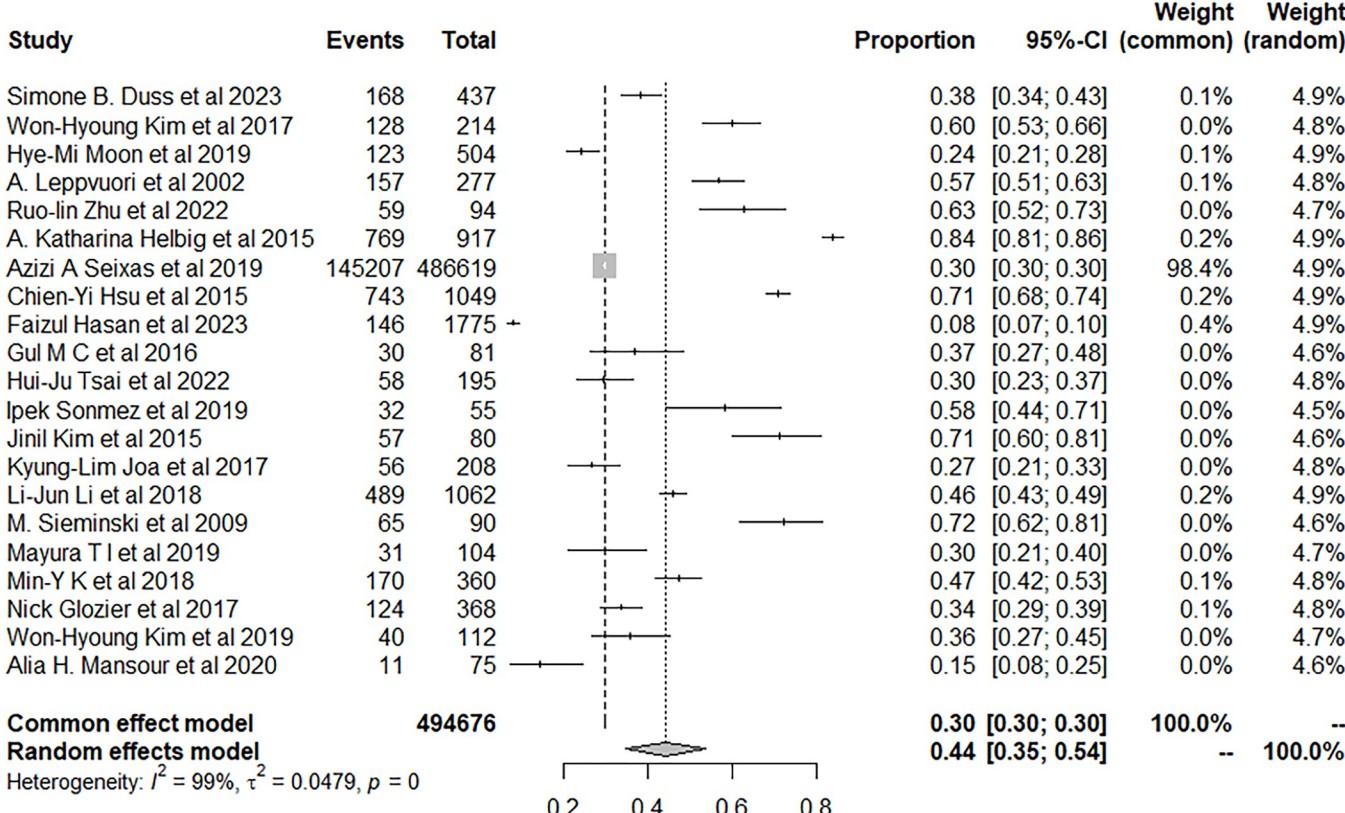

**Fig 7. Forest plot of the meta-analysis of prevalence of insomnia among non-acute stroke patients.**

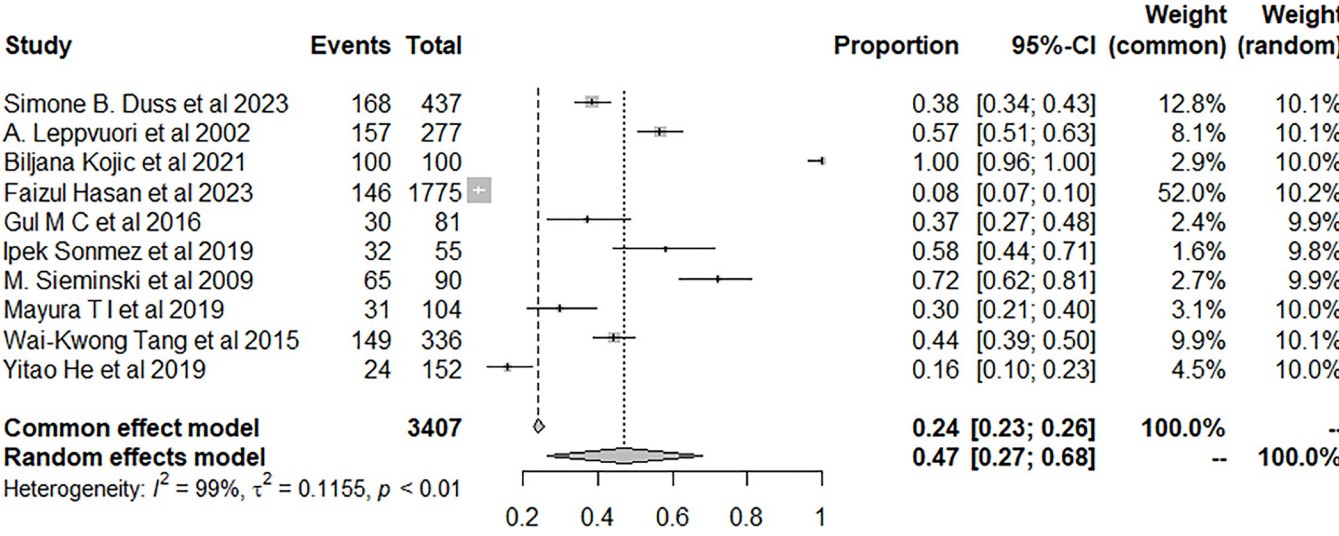

**Fig 8. Forest plot of the meta-analysis of prevalence of insomnia in patients of mean age ≥65 years with stroke patients.**

with patients without insomnia, the incidence of stroke was significantly higher in insomnia patients [45]. A similar meta-analysis showed that sleep duration was also associated with the risk of stroke, with a 5%–7% increase in stroke risk for every 1-h decrease in short sleep duration (RR = 1.05–1.07, 95% CI: 1.01–1.12) [46,47].

Insomnia after stroke is associated with the acute or chronic phase of stroke. In this study, we found that the rate of insomnia was higher in the acute phase of stroke (59.16%) than in the nonacute phase of stroke (36.31%). Luisa et al. found that polysomnography in acute IS patients showed poorer sleep quality was associated with sleep efficiency, sleep-onset awakening time in stroke patients [48]. Several factors usually caused insomnia in patients with stroke. Insomnia in patients with acute stroke was found to be associated with an increased risk of post-stroke psychiatric disorders [49].

Moreover, the age of patients with stroke and the duration of follow-up are also important factors influencing the rate of insomnia in patients with stroke. In the general population,

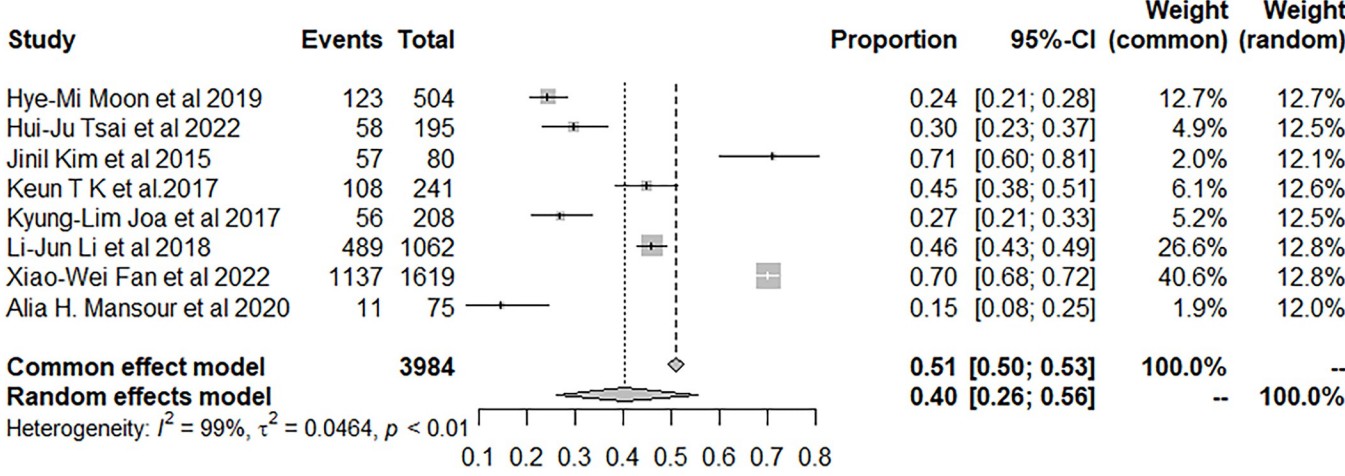

**Fig 9. Forest plot of the meta-analysis of prevalence of insomnia in patients of mean age <65 years with stroke patients.**

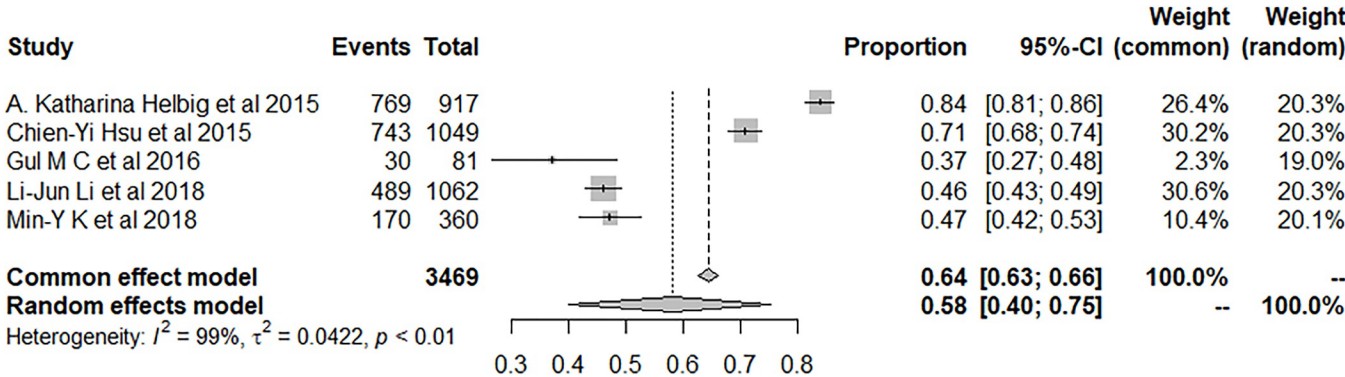

**Fig 10. Forest plot of the meta-analysis of prevalence of insomnia among patients with stroke with follow-up for ≥ 3years.**

insomnia may increase with age [50]. Studies showed a significantly higher prevalence of insomnia in elderly people [51]. Nick Glozier et al. found that the prevalence of insomnia was 16% after 6 months of stroke and 23% after 12 months of stroke [38]. The aforementioned study suggested that older patients with stroke might have an increased likelihood of experiencing insomnia during the follow-up period, and this likelihood seems to grow over time.

Insomnia assessment and diagnostic tool is also one of the factors affecting the rate of insomnia. This study found that the prevalence of insomnia using the Insomnia Assessment Diagnostic Tool was 51.16%, which was higher than the prevalence of self-reported insomnia (37.58%). In contrast, in study using the insomnia assessment and diagnostic tool, the prevalence of insomnia was different in acute phase and subacute phase stroke (32.5% vs 34.8%), whereas the overall prevalence of self-assessed insomnia also was different in acute phase and subacute phase stroke (47.1% vs 50.4%) [52]. Further large-sample studies are needed to validate these findings.

This study had some limitations. First, the study quality was not an exclusion criterion, which might have contributed to the differences in the prevalence of insomnia after stroke.

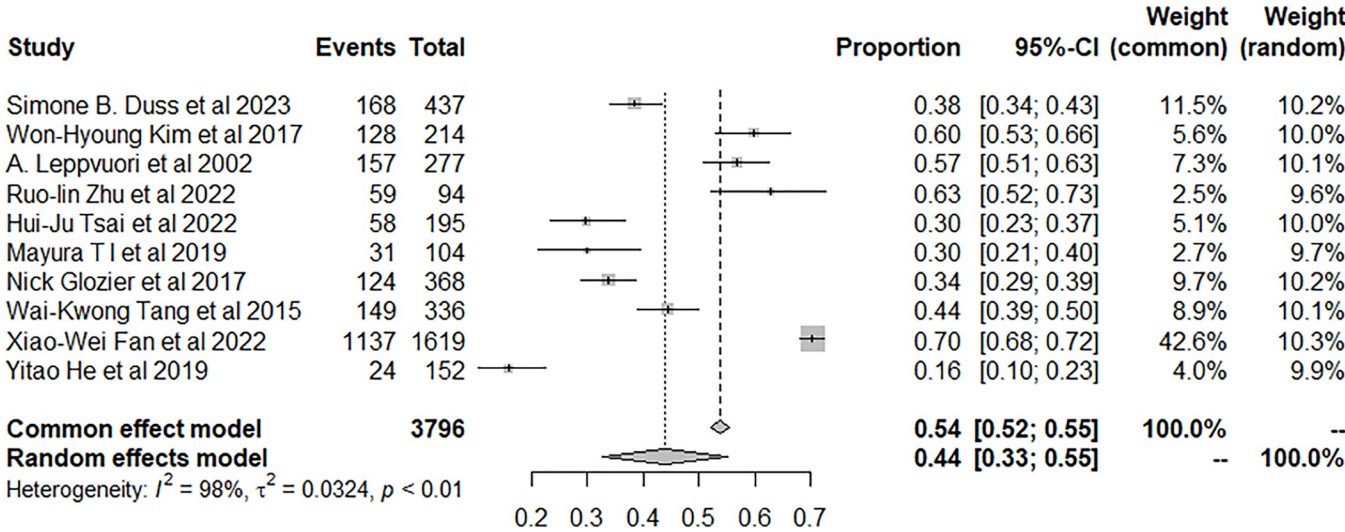

**Fig 11. Forest plot of the meta-analysis of prevalence of insomnia among patients with stroke with follow-up for < 3years.**

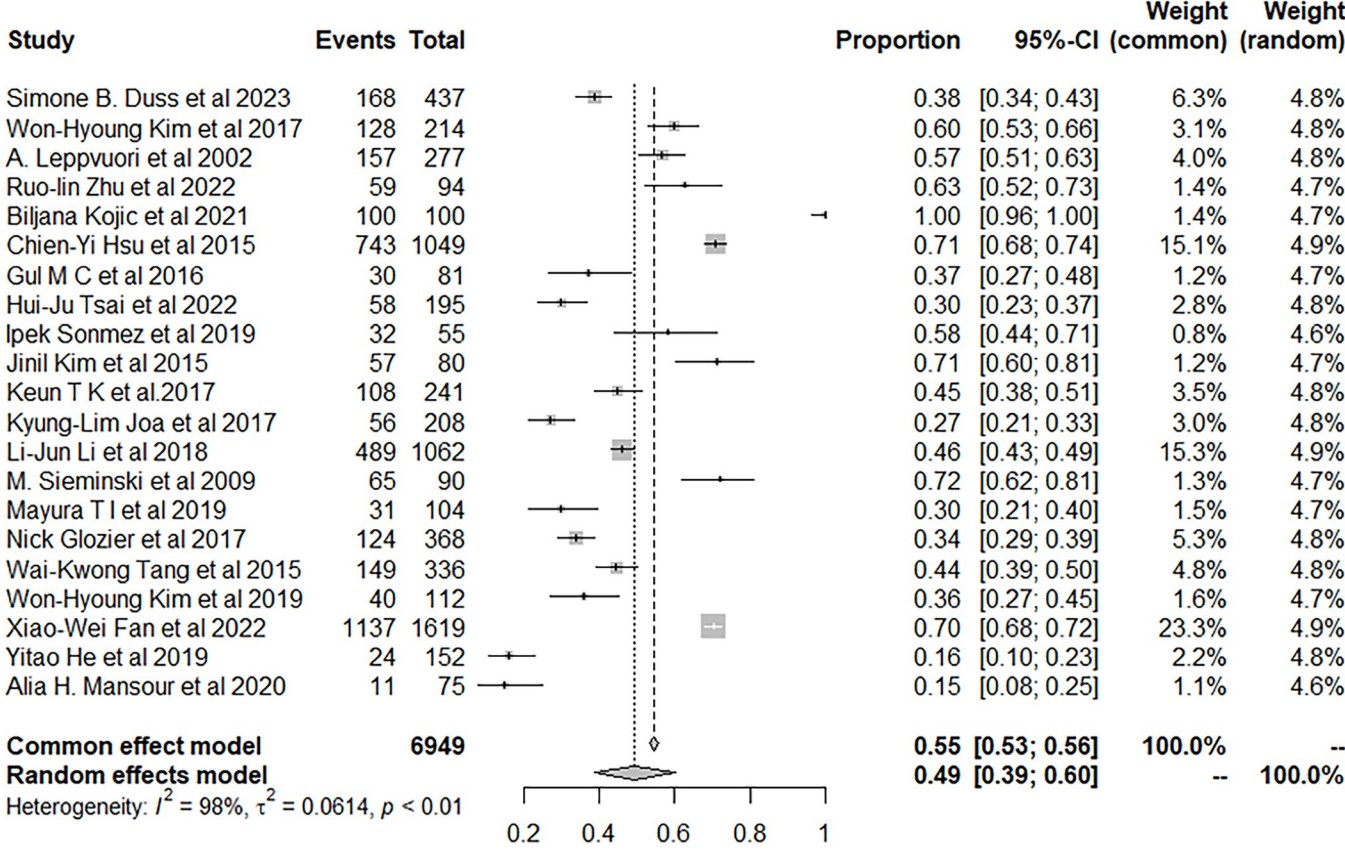

**Fig 12. Forest plot of the subgroup analysis with assessment tool.**

Studies used different tools for assessing and diagnosing insomnia, which might also have led to biased conclusions. Second, we did not study the treatment of patients with stroke and its effect on the development of insomnia.

## 5 Conclusions

Stroke may be a predisposing factor for insomnia. Insomnia is more likely to occur in acute-phase stroke, and the prevalence of insomnia increases with patient age and follow-up.

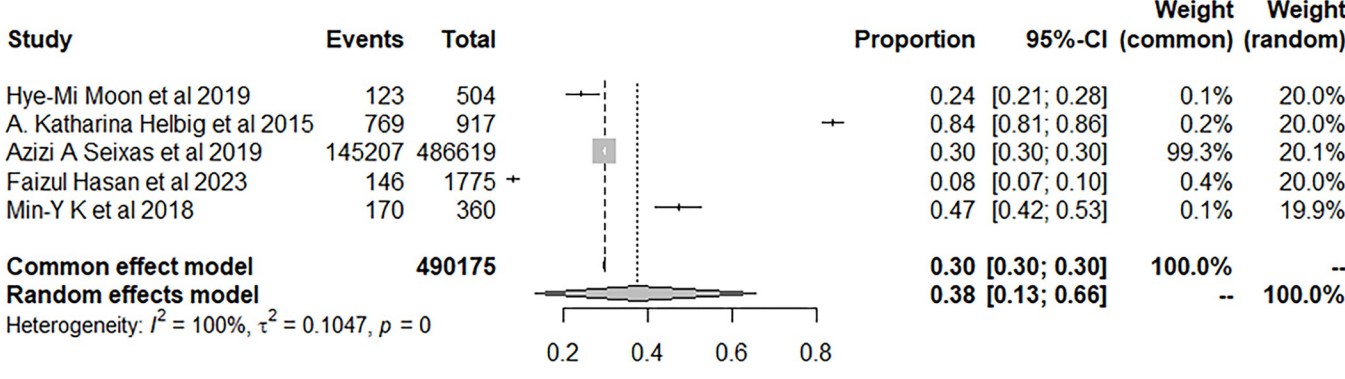

**Fig 13. Forest plot of the subgroup analysis without assessment tool.**

Further, the rate of insomnia is higher in patients with stroke who use the Insomnia Assessment Diagnostic Tool.

## Supporting information

**S1 Checklist. PRISMA 2020 checklist.**
(DOCX)

**S1 Data.**
(XLSX)

**S1 File.**
(PDF)

## Author Contributions

**Conceptualization:** Junwei Yang.

**Data curation:** Aitao Lin, Qingjing Tan, Weihua Dou, Jinyu Wu, Yang Zhang, Haohai Lin, Baoping Wei.

**Funding acquisition:** Jiemin Huang.

**Methodology:** Weihua Dou, Jinyu Wu, Yang Zhang, Haohai Lin, Baoping Wei.

**Project administration:** Juanjuan Xie.

**Supervision:** Jiemin Huang, Juanjuan Xie.

**Writing – original draft:** Qingjing Tan.

**Writing – review & editing:** Jiemin Huang.

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
