## [Decision Letter · Decision Letter 0]

3 Nov 2023

PONE-D-23-27856Development of Insomnia in patients with Stroke: A systematic meta-analysisPLOS ONE

Dear Dr. Lin,

Thank you for submitting your manuscript to PLOS ONE. After careful consideration, we feel that it has merit but does not fully meet PLOS ONE’s publication criteria as it currently stands. Therefore, we invite you to submit a revised version of the manuscript that addresses the points raised during the review process.

Based on the reviewers' suggestions, the paper needs major revision.  The reviewers' comments can be found below.

We look forward to receiving your revised manuscript.

Kind regards,

Tanja Grubić Kezele, Ph.D., M.D.

Academic Editor

PLOS ONE

Journal Requirements:

Lin A, Tan Y, Chen J, Liu X, Wu J. Development of ankylosing spondylitis in patients with ulcerative colitis: A systematic meta-analysis. PLoS One. 2023 Aug 1;18(8):e0289021. doi: 10.1371/journal.pone.0289021. PMID: 37527250; PMCID: PMC10393153.

In your revision ensure you cite all your sources (including your own works), and quote or rephrase any duplicated text outside the methods section. Further consideration is dependent on these concerns being addressed.

5. Please amend the manuscript submission data (via Edit Submission) to include authors Dr. Qingjing TAN, Dr. Weihua DOU, Dr. Jinyu WU, Dr. Yang ZHANG, Dr. Haohai LIN, Dr. Baoping WEI, and Dr. Jiemin HUANG.

6. Please clarify all figures in your manuscript and separate supporting information tables file.

Reviewers' comments:

Reviewer's Responses to Questions

**Comments to the Author**

1. Is the manuscript technically sound, and do the data support the conclusions?

Reviewer #1: Partly

Reviewer #2: Partly

Reviewer #3: Partly

2. Has the statistical analysis been performed appropriately and rigorously? 

Reviewer #1: I Don't Know

Reviewer #2: Yes

Reviewer #3: I Don't Know

3. Have the authors made all data underlying the findings in their manuscript fully available?

Reviewer #1: Yes

Reviewer #2: Yes

Reviewer #3: Yes

4. Is the manuscript presented in an intelligible fashion and written in standard English?

Reviewer #1: Yes

Reviewer #2: Yes

Reviewer #3: Yes

5. Review Comments to the Author

Reviewer #1: This is an interesting topic. Thank you for the opportunity to review your paper. I hope you find my comments helpful.

Title

The PRISMA guideline recommends that the title include systematic review and meta-analysis.

Abstract

What do you mean by year of establishment?

“A forest plot was used to determine the prevalence of insomnia among patients with stroke.” A forest plot can not determine the prevalence, a forest plot is just a way to illustrate a result.

Could you mention in the abstract, How many people screened the studies and which method was used to conduct the meta-analysis?

“Meanwhile, after follow-up for <3 years, the incidence of insomnia among patients with stroke was 43.83% (95% CI: 32.75–55.22), which was significant.” What does a significant incidence mean? This does not make sense to me.

There was no mention of difference by age in the result section, however, your conclusion mentioned that “the prevalence of insomnia increases with patient age and follow-up time.”

Introduction

This study is a replication or an update of the Baylan et al. study (Incidence and prevalence of post-stroke insomnia: A systematic review and meta-analysis) and should have discussed it upfront.

Method

Could you provide a complete search strategy? At least for PubMed. It is better than listing a few terms + etc.

Could you please list all inclusion and exclusion criteria?

“In the study, we included the cohort studies and cross-sectional studies about stroke patients who developed insomnia in English language.”

What is the rationale for including cross-sectional studies and not case-control studies for example?

“We excluded the duplicate records, case reports, reviews and so on.” What is so on?

Discussion

“First, the study quality was not an exclusion criterion, which might have contributed to the differences in the prevalence of insomnia after stroke.” What is the rationale for not excluding studies with poor quality?

About 10 studies were identified after the Baylan et al. systematic review and meta-analysis. It will be good to discuss why your study finding is different from Baylan et al. in light of these studies.

Overall:

Insomnia in patients with stroke should be differentiated from post-stroke insomnia.

Likewise, studies using diagnostic assessment tools should be differentiated from studies using non-diagnostic assessment tools.

Reviewer #2: An important research article that discuss the prevalence of insomnia post stroke , with a metanalysis and systematic review. results determined that 48.37% of patients had stroke, with increase percentage among those older than 65 years . However herein some comments to strengthen the authors' argument.

Major flaws ;

1-Authors included some studies in which the included sample patients were not all a stroke patients , and also this was mentioned in the result section of the abstract " Twenty-five studies met the inclusion criteria for

meta-analysis, with 1,193,584 participants, of which 497,049 were

patients with stroke". it is better to determine the prevalence of insomnia among stroke patients only, and not to compare with another population .

2- Polysomnographic based studies were not included entirely and herein an example of a research that was not included " Polysomnographic Characteristics of Sleep in Stroke: A Systematic Review and Meta-Analysis

Chiara Baglioni , Christoph Nissen , Adrian Schweinoch , Dieter Riemann , Kai Spiegelhalder , Mathias Berger ,

Cornelius Weiller , Annette Sterr ( 2016) and was published in PLOS one journal .

3- One polysomnographic study was mentioned , however the polysomnography device specifications and reference values were not mentioned .

4- A study about post stroke sleep disorders from Egypt was not included " Post-stroke sleep disorders in Egyptian patients by using simply administered questionnaires: a study from Ain Shams University.Alia H. Mansour, Maged Ayad, Naglaa El-Khayat, Ahmed El Sadek & Taha K. Alloush . The Egyptian Journal of Neurology, Psychiatry and Neurosurgery volume 56, Article number: 13 (2020) .

Minor Flaws ;

1- Discussion needs to be more rich and longer .

Regarding references : in some references author's first and last name are written in full , and in others only initials included . Reference 15, and 19 need revision .

Reviewer #3: The authors provide a meta analysis of the association of insomnia and stroke. They find that insomnia is common after stroke. The paper is generally well written. Specific comments follow.

The biggest issue with this paper is that one study dominates the meta analysis. Is it appropriate to do a meta analysis under such circumstances? Aren’t your results just a duplication of the dominant study? You seem to deal with this by the random effects model. Please say more about this and how it is applied to this study.

There is significant heterogeneity in the studies. Again, doesn’t this suggest that a meta analysis is not appropriate?

Your study finds associations which are not necessarily causal. Please change the language throughout the paper to reflect this association.

It’s a bit unusual to include anything other than clinical trials and cohort studies in meta analyses. How do you results change when you exclude the other study types?

6. PLOS authors have the option to publish the peer review history of their article (what does this mean?). If published, this will include your full peer review and any attached files.

Reviewer #1: No

Reviewer #2: **Yes: **Alia H. Mansour

Reviewer #3: No

---

## [Author Response · Author response to Decision Letter 0]

23 Nov 2023

Response Letter

Re: Responses to Editors and Reviewers. Manuscript Number: PONE-D-23-27856, submitted to PLOS ONE Journal, “Development of Insomnia in patients with Stroke: A systematic meta-analysis”.

Dear Editors and Reviewers,

We appreciate the detailed and constructive comments provided by you. We have carefully revised the manuscript by incorporating all the suggestions by the editors and Reviewers panel.

In the manuscript, we have tried our best to improve the writing of all text. We believe that this revised manuscript has addressed your concerns. Thank you for your help. We look forward to hearing from you about further status of our manuscript.

Kind regards,

Lin aitao

Reviewer#1:

This is an interesting topic. Thank you for the opportunity to review your paper. I hope you find my comments helpful.

Comments 1:

Title

The PRISMA guideline recommends that the title include systematic review and meta-analysis.

Author response 1: Thank you for the comment. We agreed with the reviewer. We have made corresponding revisions in the manuscript on page 1, lines 1 - 2. Modifications are highlighted with red color.

Comments 2:

Abstract

What do you mean by year of establishment?

Author response 2: Thank you for the comment. The year of establishment is about the databases start running which is in order to collecte the literatures related to the occurrence of developmental insomnia in stroke patients. We have made corresponding revisions in the manuscript on page 2, lines 31-32.

“A forest plot was used to determine the prevalence of insomnia among patients with stroke.” A forest plot can not determine the prevalence, a forest plot is just a way to illustrate a result.

Author response 2: Thank you for the comment. We agreed with the reviewer. We have made corresponding revisions in the manuscript on page 2, lines 31-35.

Could you mention in the abstract, How many people screened the studies and which method was used to conduct the meta-analysis?

Author response 2: Thank you for the comment. We showed the results of screened the studies in the abstract, and the methods of the meta-analysis were showed on pages 2, lines 31-38.

“Meanwhile, after follow-up for <3 years, the incidence of insomnia among patients with stroke was 43.83% (95% CI: 32.75–55.22), which was significant.” What does a significant incidence mean? This does not make sense to me.

Author response 2: Thank you for the comment. We have made corresponding revisions in the manuscript on pages 3, lines 51-53. Modifications are highlighted with red color.

There was no mention of difference by age in the result section, however, your conclusion mentioned that “the prevalence of insomnia increases with patient age and follow-up time.”

Author response 2: Thank you for the comment. “Similarly, the incidence of insomnia was significantly higher in the patients with stroke at a mean age of ≥65 than patients with stroke at a mean age of <65 years (47.18% vs 40.50%, P < 0.05).” was mentioned on page 3, lines 48-50.

Comments 3:

Introduction

This study is a replication or an update of the Baylan et al. study (Incidence and prevalence of post-stroke insomnia: A systematic review and meta-analysis) and should have discussed it upfront.

Author response 3: Thank you for the comment. The study was an update of the incidence of insomnia among patients with stroke. We revealed the incidence of insomnia among stroke patients from the types of stroke, the mean age of stroke patients, and the follow-up time.

Comments 4:

Method

Could you provide a complete search strategy? At least for PubMed. It is better than listing a few terms + etc.

Author response 4: Thank you for the comment. We provided a complete search strategy for the PubMed database which showed in Fig 1.

Could you please list all inclusion and exclusion criteria?

Author response 4: Thank you for the comment. The inclusion and exclusion criteria were listed on page 5-6, lines 100-114.

“In the study, we included the cohort studies and cross-sectional studies about stroke patients who developed insomnia in English language.”

What is the rationale for including cross-sectional studies and not case-control studies for example?

Author response 4: Thank you for the comment. The study revealed the occurrence of insomnia during stroke follow-up. We included the cohort studies and cross-sectional studies and not case-control studies, because most of cohort studies and cross-sectional studies were large sample, multi-center, and long follow-up period, while the follow-up period of the case-control study was short.

“We excluded the duplicate records, case reports, reviews and so on.” What is so on?

Author response 4: Thank you for the comment. We have made corresponding revisions in the manuscript on pages 6, lines 111-114. Modifications are highlighted with red color.

Comments 5:

Discussion

“First, the study quality was not an exclusion criterion, which might have contributed to the differences in the prevalence of insomnia after stroke.” What is the rationale for not excluding studies with poor quality?

Author response 5: Thank you for the comment. In the study, 26 studies from 11 countries were included. The methodological quality of the included studies was assessed using the Critical Appraisal Tool for Prevalence Studies, and the details were shown in Table 2 which most of the studies were medium to high quality.

About 10 studies were identified after the Baylan et al. systematic review and meta-analysis. It will be good to discuss why your study finding is different from Baylan et al. in light of these studies.

Author response 5: Thank you for the comment. In the study, we included more studies about the prevalence of insomnia in patients with stroke. We revealed the incidence of insomnia among stroke patients from the types of stroke, the mean age of stroke patients, and the follow-up time. This is different from the Baylan et al. systematic review and meta-analysis.

Overall:

Insomnia in patients with stroke should be differentiated from post-stroke insomnia.

Likewise, studies using diagnostic assessment tools should be differentiated from studies using non-diagnostic assessment tools.

Author response 5: Thank you for the comment. We will make corresponding revisions in the manuscript.

Reviewer #2: An important research article that discuss the prevalence of insomnia post stroke, with a metanalysis and systematic review. results determined that 48.37% of patients had stroke, with increase percentage among those older than 65 years. However herein some comments to strengthen the authors' argument.

Major flaws:

1-Authors included some studies in which the included sample patients were not all a stroke patients, and also this was mentioned in the result section of the abstract " Twenty-five studies met the inclusion criteria for

meta-analysis, with 1,193,584 participants, of which 497,049 were

patients with stroke". it is better to determine the prevalence of insomnia among stroke patients only, and not to compare with another population .

Author response 1: Thank you for the comment. In the study, we included the cohort studies and cross-sectional studies to reveal the incidence of insomnia among patients with stroke. While some studies were cross-sectional Population based in which the included sample patients were not all a stroke patients, but we just revealed the incidence of insomnia among stroke patients. The details were shown in Table 1.

2- Polysomnographic based studies were not included entirely and herein an example of a research that was not included " Polysomnographic Characteristics of Sleep in Stroke: A Systematic Review and Meta-Analysis

Chiara Baglioni , Christoph Nissen , Adrian Schweinoch , Dieter Riemann , Kai Spiegelhalder , Mathias Berger ,

Cornelius Weiller , Annette Sterr ( 2016) and was published in PLOS one journal.

Author response 2: Thank you for the comment. In the study, we revealed the occurrence of developmental insomnia in stroke patients by the subject terms, such as “Stroke”, “Cerebrovascular Accident”, “Insomnia”, “Insomnia Disorder”, etc. And in the included studies, we revealed the incidence of insomnia among stroke patients from the types of stroke, the mean age of stroke patients, the follow-up time and the use of insomnia assessment diagnostic tools or not. We did not conducted a systematic review and meta-analysis of polysomnographic studies comparing stroke to control populations.

3-One polysomnographic study was mentioned, however the polysomnography device specifications and reference values were not mentioned.

Author response 3: Thank you for the comment. In the study, the subgroup analyse was performed based on the use of insomnia assessment diagnostic tools (clinical assessment diagnostic tools vs self-report), so we just compared the different prevalence of insomnia among stroke patients who used the insomnia assessment tools or not.

4- A study about post stroke sleep disorders from Egypt was not included "Post-stroke sleep disorders in Egyptian patients by using simply administered questionnaires: a study from Ain Shams University.Alia H. Mansour, Maged Ayad, Naglaa El-Khayat, Ahmed El Sadek & Taha K. Alloush . The Egyptian Journal of Neurology, Psychiatry and Neurosurgery volume 56, Article number: 13 (2020).

Author response 4: Thank you for the comment. We have made corresponding revisions in the manuscript.

5- Discussion needs to be more rich and longer.

Author response 5: Thank you for the comment. We have made corresponding revisions in the manuscript.

6-Regarding references : in some references author's first and last name are written in full , and in others only initials included . Reference 15, and 19 need revision .

Author response 6: Thank you for the comment. We have made corresponding revisions in the manuscript.

Reviewer #3: The authors provide a meta analysis of the association of insomnia and stroke. They find that insomnia is common after stroke. The paper is generally well written. Specific comments follow.

Comments 1:

The biggest issue with this paper is that one study dominates the meta analysis. Is it appropriate to do a meta analysis under such circumstances? Aren’t your results just a duplication of the dominant study? You seem to deal with this by the random effects model. Please say more about this and how it is applied to this study.

Author response 1: Thank you for the comment. In the study, 26 studies were included in the meta-analysis. Although one study included a relatively large number of stroke, but we also compared the different prevalence of insomnia among stroke patients by follow-up time, age, and the insomnia assessment tools or not. The frequency of insomnia development in stroke during follow-up was detected by a fixed or random-effects model. If P ≥ 0.10 and I2 ≤ 50%, the data were homogeneous across studies and were analysed using a fixed effects model. If P< 0.10 and/or I2 > 50%, we need to find the source of heterogeneity, or in cases where the source of heterogeneity was unclear, a random-effects model was used.

Comments 2:

There is significant heterogeneity in the studies. Again, doesn’t this suggest that a meta analysis is not appropriate?

Author response 2: Thank you for the comment. The heterogeneity in the study did not mean the meta analysis is not appropriate. Conversely, we should analysis the prevalence of insomnia among stroke patients in subgroup analysis.

Comments 3:

Your study finds associations which are not necessarily causal. Please change the language throughout the paper to reflect this association.

Author response 3: Thank you for the comment. We have made corresponding revisions in the manuscript.

Comments 4:

It’s a bit unusual to include anything other than clinical trials and cohort studies in meta analyses. How do you results change when you exclude the other study types?

Author response 4: Thank you for the comment. In the study, we included the cohort studies and cross-sectional studies to reveal the incidence of insomnia among patients with stroke. This is a cross-sectional study.

---

## [Decision Letter · Decision Letter 1]

12 Dec 2023

PONE-D-23-27856R1Development of Insomnia in patients with Stroke: A systematic review and meta-analysisPLOS ONE

Dear Dr. Lin,

Thank you for submitting your manuscript to PLOS ONE. After careful consideration, we feel that it has merit but does not fully meet PLOS ONE’s publication criteria as it currently stands. Therefore, we invite you to submit a revised version of the manuscript that addresses the points raised during the review process.

Based on the reviewers' suggestions, the paper needs major revision.  The reviewers' comments can be found below.

We look forward to receiving your revised manuscript.

Kind regards,

Tanja Grubić Kezele, Ph.D., M.D.

Academic Editor

PLOS ONE

Reviewers' comments:

Reviewer's Responses to Questions

**Comments to the Author**

1. If the authors have adequately addressed your comments raised in a previous round of review and you feel that this manuscript is now acceptable for publication, you may indicate that here to bypass the “Comments to the Author” section, enter your conflict of interest statement in the “Confidential to Editor” section, and submit your "Accept" recommendation.

Reviewer #2: All comments have been addressed

Reviewer #3: All comments have been addressed

2. Is the manuscript technically sound, and do the data support the conclusions?

Reviewer #2: Partly

Reviewer #3: Yes

3. Has the statistical analysis been performed appropriately and rigorously? 

Reviewer #2: No

Reviewer #3: Yes

4. Have the authors made all data underlying the findings in their manuscript fully available?

Reviewer #2: Yes

Reviewer #3: Yes

5. Is the manuscript presented in an intelligible fashion and written in standard English?

Reviewer #2: Yes

Reviewer #3: Yes

6. Review Comments to the Author

Reviewer #2: Thank you for this work and meta analysis on insomnia with stroke . Thank you for the responses you have made .

1- in line 105 under eligibility criteria and study selection , Epwarth sleepiness scale is written as an assessment scale for insomnia however it is an assessment scale for EDS ( excessive day time sleepiness )

2- the comparison between the percentage of insomnia in Ischemic stroke and haemorrhage with P value was not mentioned .

3- It would be better to correlate insomnia with other factors in stroke patients as DM , and HTN as this might contribute to the insomnia , also correlation with the size and site of the stroke will enrich the manuscript and help in reaching to the conclusion written.

Reviewer #3: I have no additional comments for the authors.

7. PLOS authors have the option to publish the peer review history of their article (what does this mean?). If published, this will include your full peer review and any attached files.

Reviewer #2: No

Reviewer #3: No

---

## [Author Response · Author response to Decision Letter 1]

3 Jan 2024

Response Letter

Re: Responses to Reviewers. Manuscript Number: PONE-D-23-27856, submitted to PLOS ONE Journal, “Development of Insomnia in patients with Stroke: A systematic meta-analysis”.

Dear Editors and Reviewers,

We appreciate the detailed and constructive comments provided by you. We have carefully revised the manuscript by incorporating all the suggestions by the editors and Reviewers panel.

In the manuscript, we have tried our best to improve the writing of all text. We believe that this revised manuscript has addressed your concerns. Thank you for your help. We look forward to hearing from you about further status of our manuscript.

Kind regards,

Lin aitao

Reviewer#2:

Comments 1:

in line 105 under eligibility criteria and study selection , Epwarth sleepiness scale is written as an assessment scale for insomnia however it is an assessment scale for EDS (excessive day time sleepiness).

Author response 1: Thank you for the comment. We agreed with the reviewer. We have made corresponding revisions in the manuscript on page 5, lines 97 - 98.

Comments 2:

the comparison between the percentage of insomnia in Ischemic stroke and haemorrhage with P value was not mentioned.

Author response 2: Thank you for the comment. We regretted that we did not get the prevalence of insomnia in Ischemic stroke and haemorrhage, respectively. So, in the study, the comparison between the percentage of insomnia in Ischemic stroke and haemorrhage with P value was not mentioned. More research needs to be investigated in the future.

Comments 3:

It would be better to correlate insomnia with other factors in stroke patients as DM, and HTN as this might contribute to the insomnia, also correlation with the size and site of the stroke will enrich the manuscript and help in reaching to the conclusion written.

Author response 3: Thank you for the comment. We agreed with the reviewer. But in the study, 26 studies were included in the meta-analysis. We regretted that we did not get the correlate insomnia with other factors in stroke patients. Besides, we did not get the data of the size and site of the stroke patients, too. More research needs to be investigated in the future.

---

## [Decision Letter · Decision Letter 2]

16 Jan 2024

Development of Insomnia in patients with Stroke: A systematic review and meta-analysis

PONE-D-23-27856R2

Dear Dr. Lin,

We’re pleased to inform you that your manuscript has been judged scientifically suitable for publication and will be formally accepted for publication once it meets all outstanding technical requirements.

Kind regards,

Tanja Grubić Kezele, Ph.D., M.D.

Academic Editor

PLOS ONE

Additional Editor Comments (optional):

Reviewers' comments:

Reviewer's Responses to Questions

**Comments to the Author**

1. If the authors have adequately addressed your comments raised in a previous round of review and you feel that this manuscript is now acceptable for publication, you may indicate that here to bypass the “Comments to the Author” section, enter your conflict of interest statement in the “Confidential to Editor” section, and submit your "Accept" recommendation.

Reviewer #2: All comments have been addressed

Reviewer #3: All comments have been addressed

2. Is the manuscript technically sound, and do the data support the conclusions?

Reviewer #2: Yes

Reviewer #3: Yes

3. Has the statistical analysis been performed appropriately and rigorously? 

Reviewer #2: I Don't Know

Reviewer #3: Yes

4. Have the authors made all data underlying the findings in their manuscript fully available?

Reviewer #2: No

Reviewer #3: Yes

5. Is the manuscript presented in an intelligible fashion and written in standard English?

Reviewer #2: Yes

Reviewer #3: Yes

6. Review Comments to the Author

Reviewer #2: a meta analysis of post stroke insomnia which is an important factor in morbidity and burden after stroke , and concluded that the use of insomnia quantitative questionnaires are important . And that insomnia increase with increasing age . Clinical application to this meta analysis is that it is important to manage insomnia in the acute phase , which will improve quality of life and contribute to recovery .

Reviewer #3: None

7. PLOS authors have the option to publish the peer review history of their article (what does this mean?). If published, this will include your full peer review and any attached files.

Reviewer #2: No

Reviewer #3: No

---

## [Editor Report · Acceptance letter]

27 Mar 2024

PONE-D-23-27856R2 

PLOS ONE

Dear Dr. Lin, 

I'm pleased to inform you that your manuscript has been deemed suitable for publication in PLOS ONE. Congratulations! Your manuscript is now being handed over to our production team.

Kind regards, 

on behalf of

Prof. dr. Tanja Grubić Kezele 

Academic Editor

PLOS ONE